# Incorporating male sterility increases hybrid maize yield in low input African farming systems

Sarah Collinson [1], Esnath Hamdziripi [2], Hugo De Groote [3], Michael Ndegwa [3], Jill E. Cairns [2], Marc Albertsen [4], Dickson Ligeyo[5], Kingstone Mashingaidze[6] & Michael S. Olsen [3✉]

Maize is a staple crop in sub-Saharan Africa, but yields remain sub-optimal. Improved breeding and seed systems are vital to increase productivity. We describe a hybrid seed production technology that will benefit seed companies and farmers. This technology improves efficiency and integrity of seed production by removing the need for detasseling. The resulting hybrids segregate 1:1 for pollen production, conserving resources for grain production and conferring a 200 kg ha$^{-1}$ benefit across a range of yield levels. This represents a 10% increase for farmers operating at national average yield levels in sub-Saharan Africa. The yield benefit provided by fifty-percent non-pollen producing hybrids is the first example of a single gene technology in maize conferring a yield increase of this magnitude under low-input smallholder farmer conditions and across an array of hybrid backgrounds. Benefits to seed companies will provide incentives to improve smallholder farmer access to higher quality seed. Demonstrated farmer preference for these hybrids will help drive their adoption.

[1] Corteva Agriscience, Woodland, CA, USA. [2] International Maize and Wheat Improvement Centre (CIMMYT), Harare, Zimbabwe. [3] International Maize and Wheat Improvement Centre (CIMMYT), Nairobi, Kenya. [4] Corteva Agriscience, Johnston, IA, USA. [5] Kenya Agricultural and Livestock Research Organization (KALRO), Kitale, Kenya. [6] Agricultural Research Council (ARC), Potchefstroom, South Africa. ✉email: M.Olsen@cgiar.org

Increasing the productivity of smallholder farmers in sub-Saharan Africa (SSA) is an important step toward improving livelihoods and reducing risk[1]. Maize yields in SSA remain the lowest in the world, and historical production increases are associated with an unsustainable increase in maize area. At current yield levels, the area under maize cultivation must increase by 184% to meet future food security needs[2]. Obsolete varieties that were developed for climate conditions that have subsequently changed are still widely grown. Rapid-cycle breeding and faster varietal replacement are essential to increasing yields under changing climates[3].

Progress in SSA has been made through modernizing breeding programs, engagement with seed companies, and the development and delivery of elite, stress-tolerant varieties[4]. Improved maize production in Ethiopia, through improved maize genetics and other agronomic inputs[5], has helped lift an estimated 788,000 people out of poverty annually[6]. There is an increasing focus on improving the efficiency of public sector maize breeding[7] but seed production remains a key bottleneck in SSA[8].

Hybrids are maize varieties in which the seed is produced by crossing two different parent lines, increasing the yield through heterosis. Detasseling in hybrid seed production in SSA is manual, unlike in other regions of the world, leading to higher cost of the seed and issues with quality[9,10]. Most commercially available hybrids in SSA are three-way hybrids which are formed by crossing two lines together to form a single cross female parent and then crossing the single cross female to a third inbred to produce commercial seed. Three-way cross hybrids are common in SSA, even if yields are lower since the cost of goods sold (COGS) is lower due to the higher seed yield of single cross females compared with inbred lines. Technologies to reduce both COGS and the complexity of producing high-quality hybrids would offer smaller seed companies greater opportunities to provide new hybrids to smallholder farmers[11]. Seed production technology (SPT) is a process previously used by Corteva Agriscience to produce commercial hybrid maize seeds in the United States. The original SPT system was based on a recessive male sterility gene, *ms45*, and it utilized a transgenic maintainer cassette containing the dominant Ms45 allele to restore fertility to *ms45* homozygous plants, an α-amylase gene to render transgenic pollen non-viable, and a seed color marker gene[12]. SPT enables the production of homozygous male sterile non-transgenic seed.

Subsequent development of an SPT system based on the dominant male sterility gene, *Ms44*, enables seed increase of homozygous dominant non-pollen producing (NPP) inbred and heterozygous NPP female single-cross parent plants[13]. The *Ms44*-SPT system is well suited for three-way hybrid production as it eliminates the need for detasseling maize hybrid production fields during both hybridization steps. Three-way hybrids produced using heterozygous NPP female parents segregate 1:1 pollen-producing (PP) and NPP and have been shown to increase yield by 8.5% under ultra-low nitrogen managed stress field testing in the US[13]. Such hybrids are designated 50% non-pollen-producing (FNP). An alternative dominant male sterility (DMS) system has subsequently been developed using the PHD-finger transcription factor *ZmMs7* together with a transgenic restoration system[14]. This system is hypothesized to deliver a similar yield benefit to the Ms44-based system in an FNP hybrid form although field validation has not yet been reported.

Biotechnology has had mixed success from a scaling perspective[15]. Translating results under controlled stress environments in experimental research stations to on-farm, with a repeatable yield advantage across a range of environments and multiple genetic backgrounds, has proved complex even in high-yielding, uniform environments[16]. The transfer of genetic technologies developed under controlled environments in the USA to low-yielding conditions in SSA has had limited success[15]. In SSA, maize is primarily grown in challenging environments, with high spatial variation even within a single smallholder farm[17]. Technology development centered in experimental research stations may lead to products that fail to perform in the target environment. Recognizing this and the need to co-develop technological solutions with farmers is leading to an increased interest to move testing on farm[18,19]. Although FNP hybrids have demonstrated yield benefits in limited testing in the US, yield levels of these trials (6.3–8.1 Mg ha$^{-1}$) were 3–4 times higher than those typically found in farmers' fields in SSA. The impact of the FNP trait on yield needs to be measured robustly across diverse on-farm sites, environments, and genetic backgrounds in order to assess the potential for FNP hybrids in SSA. Low fertilizer use (<17 kg ha$^{-1}$) is a major factor contributing to the yield gap in SSA[20], particularly in female-managed plots[21], and is exacerbated by low and variable returns on investment[22]. Here we investigate the potential of FNP hybrids to increase maize yields under a range of conditions including low input and drought-stressed conditions commonly encountered by smallholder farmers in SSA. Agricultural research often involves only researchers, without any participation from farmers[23]. Working with the primary beneficiaries is essential to ensure an understanding of what the user needs or wants in order to facilitate adoption[10]. For this reason, trials were conducted largely on-farm with primary beneficiaries and farmer perceptions of FNP hybrids were evaluated. The primary objectives of this study were to quantify the yield difference of FNP hybrids in on-farm conditions with smallholder farmers in Africa and to assess farmer perceptions of FNP hybrids to determine the likelihood that an FNP hybrid would be accepted in the market by the primary intended beneficiaries. Additionally, we investigate changes in agronomic traits and yield components associated with the FNP trait.

## Results

**FNP hybrids have increased yield relative to PP hybrids**. Multiple hybrid pairs (FNP and PP versions) were grown in both on-station trials (OST) and on-farm field trials (OFT) across Kenya, South Africa, and Zimbabwe, from 2016 to 2019 (Supplementary Fig. 1). The FNP trait was evaluated in 26 different hybrid backgrounds. Trials were conducted in 112 locations (Supplementary Table 1). FNP hybrids demonstrate a 202 kg ha$^{-1}$ advantage over PP hybrids with equivalent grain moisture (Table 1). FNP hybrids increase yield over a broad range of environments. When averaged across all hybrids within a location, the grain yield of FNP hybrids, relative to their PP controls, was consistently higher across yield levels (Fig. 1a). FNP hybrids significantly out-yielded the PP controls in 75% of the locations tested, with an overall average yield increase of 202 kg ha$^{-1}$. Absolute yield improvement was consistent across yield levels (Fig. 1a). Predicted yield improvement is 192 kg ha$^{-1}$ (9.6%) in highly stressful, low-potential environments (2000 kg ha$^{-1}$) and 229 kg ha$^{-1}$ (2.4%) in high-potential conditions (8000 kg ha$^{-1}$) (Fig. 1b).

The yield benefit of FNP was consistent across 19 hybrid backgrounds with more than 22 locations of data (Fig. 2). Seven hybrids were not included as they were grown in 12 or fewer locations. From these 19 hybrids, the average yield advantage of FNP in single cross hybrids was 178 kg ha$^{-1}$ and for three-way crosses the benefit was 264 kg ha$^{-1}$.

In these same trials, photos of all the ears harvested from NPP and PP plants were taken at harvest, and image analysis was used to estimate ear parameters[24]. Table 1 indicates that NPP plants had a significant 5.9% increase in the number of kernels per plant and a small but significant increase in 100 kernel weight of 0.9%. Ear length was increased significantly (4.9%), reflecting the larger number of kernels for NPP plants.

**Table 1 Grain yield, yield components, plant and ear height, and tassel traits of 50% non-pollen producing (FNP) and pollen producing (PP) control hybrids.**

| Trait | Pollen producing (PP) | Fifty-percent non-pollen producing (FNP) | Difference | Change (%) | P-value | N |
|---|---|---|---|---|---|---|
| Yield (kg ha$^{-1}$) | 3916.5 ± 73.2 | 4118.6 ± 73.3 | 202.1 | 5.2 | <0.0001 | 4585 |
| Moisture (%) | 18.18 ± 0.25 | 18.22 ± 0.25 | 0.03 | 0.20 | 0.62 | 3923 |
| Ear height (m) | 1.01 ± 0.01 | 1.00 ± 0.01 | −0.01 | −1.1 | <0.01 | 471 |
| Plant height (m) | 1.93 ± 0.01 | 1.86 ± 0.01 | −0.07 | −3.8 | <0.0001 | 469 |
| Grain weight (g) | 87.6 ± 3.7 | 95.2 ± 3.7 | 7.6 | 8.7 | <0.0001 | 459 |
| Number of tassel branches | 16.7 ± 0.18 | 15.3 ± 0.18 | −1.4 | −8.5 | <0.0001 | 475 |
| Tassel weight (g) | 4.0 ± 0.09 | 3.73 ± 0.09 | −0.27 | −6.7 | <0.001 | 475 |
| Kernel number | 281.3 ± 3.7 | 297.9 ± 3.7 | 16.6 | 5.9 | <0.001 | 464 |
| Grain weight (g) | 89.6 ± 1.4 | 95.1 ± 1.4 | 5.6 | 6.2 | <0.001 | 464 |
| 100 kernel weight (g) | 31.7 ± 0.34 | 32.0 ± 0.34 | 0.3 | 0.9 | <0.01 | 469 |
| Ear length (cm) | 13.2 ± 0.18 | 13.9 ± 0.18 | 0.64 | 4.9 | <0.0001 | 469 |

Grain yield and grain moisture were measured in on-farm and on-station trials. Ear and plant height, grain weight, number of tassel branches, and tassel weight were measured on individual non-pollen producing (NPP) or PP plants tagged within an FNP hybrid plot on a subset of on-station locations. Kernel number, grain weight, 100 kernel weight, and ear length were estimated from image analysis of NPP and PP ears taken from tagged plants within an FNP hybrid plot on a subset of on-station locations.

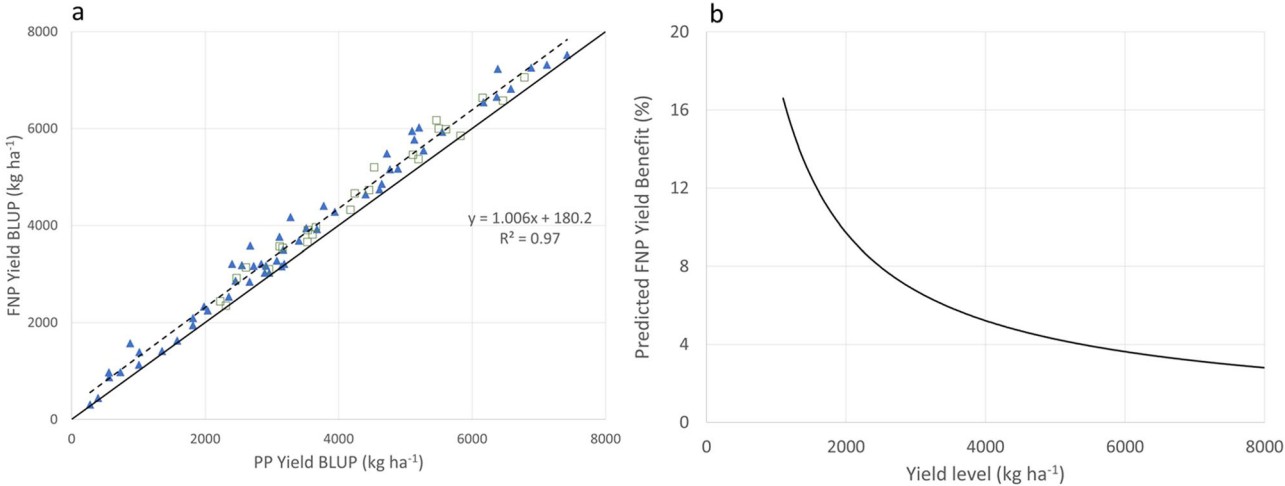

**Fig. 1 Yield benefit of 50% non-pollen producing (FNP) hybrids over conventional (PP) hybrids.** Yield (kg ha$^{-1}$) of 50% non-pollen producing (FNP) hybrids compared with conventional PP hybrids grown across multiple locations and years as shown in Supplementary Table 1 and Supplementary Fig. 1. **a** FNP hybrids yield (kg ha$^{-1}$) (y-axis) plotted against the yield of pollen-producing (PP) conventional hybrids (x-axis). Each point represents the mean of four to nine hybrid backgrounds for on-farm trials (OFT) (blue diamond) and 4–15 hybrid backgrounds for on-station trial (OST) (yellow square). The solid black line represents the 1:1 relationship and the fitted regression line is shown (blue dotted). **b** Percent yield increase predicted by growing FNP hybrids (y-axis) plotted against location mean yield (x-axis). Yield increase was projected using the fitted linear regression in **a** to predict the yield of FNP hybrids.

**Farmer participatory evaluation.** In Kenya, 2697 farmers (62% women) visited the trials to evaluate the FNP technology at eight different sites in 2017 and 2018 (Supplementary Table 2). When participants were first asked to score the importance of the different criteria on a scale of 0 (not important) to 3 (very important), they gave high scores to most of the criteria. During the mid-season evaluation, the criteria with the highest scores were yield, early maturity, ear size, and the number of ears, which all received an average score between 2.5 and 2.7. When farmers were asked if tassel formation was important during the mid-season evaluation, they scored the trait very high (2.68) second only to yield (2.69) (out of a maximum of 3). Similarly, the amount of pollen shed received an importance score of 2.6. During the mid-season evaluation in 2017, farmers scored the conventional PP hybrids significantly higher for tassel formation and pollen shed than the FNP hybrids, indicating they can clearly distinguish the two types (Table 2). In the mid-season, both the yield score and the overall score of the FNP hybrids were

significantly higher than that of the conventional, PP hybrids. Otherwise, there were few differences between the scores for the individual criteria. At the end-season (harvest) evaluation, there was no difference between scores for tassel formation of PP and FNP hybrids, indicating that participants could no longer tell the difference. At harvest, FNP hybrids generally scored better on several criteria, including significant differences in ear size and yield, and for the overall evaluation. At the mid-season evaluation in 2018, scores for the amount of pollen shed were similar between PP and FNP hybrids, the latter even getting slightly higher scores for tassel formation. During group discussions after the (individual) evaluations, farmers explained they now understood the trait and did not give FNP hybrids lower scores, even though they recognized the morphological differences. The results at harvest in 2018 were similar to those in 2017: there was no difference between the two hybrid types for tassel formation, but FNP hybrids scored higher on yield and overall evaluation compared to conventional hybrids. The results indicate that

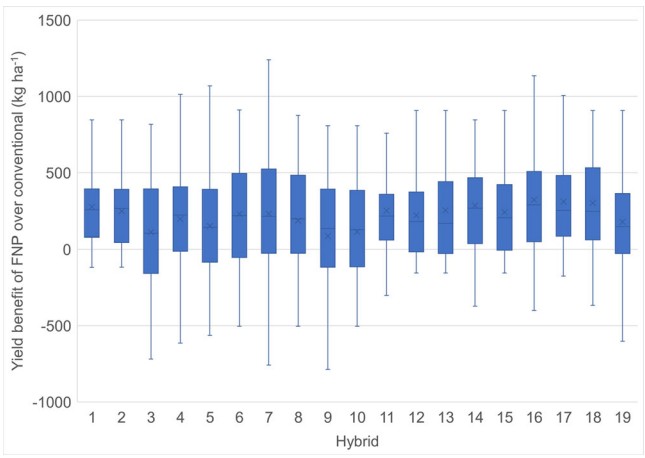

**Fig. 2 Yield benefits of FNP hybrids are consistent across multiple hybrids.** Box and whisker plots indicate the range in yield benefit (kg ha$^{-1}$) for FNP hybrid compared with conventional control for 19 hybrids, each grown in at least 20 locations. Hybrids 1–11 are single crosses, 12–19 are three-way crosses. Seven hybrids were removed as they were grown in 12 or fewer locations. $X$ represents the mean, the solid box represents 25th–75th percentile and the whiskers represent the minimum and maximum of the distribution. Outliers (>1.5 outside the outer quartile) were removed. $N$ (number of locations) are 22 (hybrids 12, 13, 15, 19), 26 (hybrids 1, 11, 14, 17), 34 (hybrids 2, 8, 9), 38 (hybrids 3, 6), 40 (hybrid 10), 48 (hybrids 16, 18), 62 (hybrid 5), 68 (hybrid 4), and 74 (hybrid 7).

farmers can distinguish FNP from PP hybrids and identify them as higher yielding and better overall (Table 2).

**Impact assessment.** At this stage, technology adoption rates are unknown but based on preliminary discussions with seed companies two scenarios can be considered. An adoption rate of FNP of 10% of the current area in maize hybrids seems a reasonable low-end scenario, while 25% would be an optimistic scenario. Based on FAO statistics and the adoption literature, the maize area in the top 25 maize-producing countries in Africa is estimated at 36.6 Mha, of which 34.2% or 12.6 Mha is planted to hybrids. The total seed needed for the low scenario (10% adoption by hybrid users) is calculated at 31,390 tonnes, and 78,000 tonnes for the high scenario (25% adoption). At an adoption rate of 10%, only 11 countries have a demand of more than 1000 tonnes (17 have a demand of >100 tonnes). The total demand for FNP seed for these 11 countries adds up to 27,906 tonnes, 93% of the total (Supplementary Table 4). To compare the benefits to the cost, we use net present value (NPV), internal rate of return (IRR), and benefit-cost ratio (BCR)[25] (Supplementary Table 5). Based on the current cost of the development of the technology, from 1 to 1.6 million $ yr$^{-1}$, the discounted cost comes to $28.9 million. For the benefits, we assume the technology to be on the market in 2023, and to take 10 years to reach the target 10% adoption (market penetration of FNP hybrids as a percentage of hybrid seed), keeping maize production constant. Under this basic scenario, maize production is expected to increase by 244,204 tonnes per year at the target adoption rate, valued at $40 million. The discounted benefits, up to 2040, are estimated at 180 M$.

The NPV is calculated at 152 M$, the BCR at 6.25, and the IRR at 24%. Under the optimistic scenario, the adoption rate of FNP reaches 25% of total hybrid use, and the extra production is estimated at 610,511 tonnes annually, valued at $100 million. Under this scenario, the discounted benefits up to 2040 are estimated at $452 million, the NPV at $423 million, the BCR at 16, and the IRR at 32%.

## Discussion

A key objective of this study was to move the evaluation of the FNP trait from US germplasm tested in managed field conditions to African germplasm tested in farmers' fields in SSA. In farmers' fields across hybrid backgrounds in Africa, we demonstrate that FNP hybrids, segregating for Ms44, increase yield by ~200 kg ha$^{-1}$ at current SSA yield levels. There is an urgent need to increase genetic gain for yield under low fertility conditions; observed rates of genetic gain under drought are 23–32 kg ha$^{-1}$ yr$^{-1}$ and low nitrogen 21 kg ha$^{-1}$ yr$^{-1}$ [26]. The yield benefit of FNP hybrids under stress conditions represents at least 6 years of progress in plant breeding. This study demonstrates the ability of FNP hybrids to deliver 10–20% yield increase under extremely stressful growing conditions faced by millions of smallholder maize farmers. The stability of the yield benefit across genetic backgrounds indicates that FNP can be successfully deployed across an array of hybrids to meet the needs of farmers in various agroecological zones throughout SSA. On-farm trials are being increasingly scrutinized due to high input and yield levels that are not representative of the actual realities of the farmers testing the technologies[27,28]. Our aim was to evaluate yield benefits at close to national average yield levels by targeting farmer-managed on-farm trials with minimal nitrogen inputs, typical of many smallholder farmers. Participatory research was a key component given the visual differences of FNP hybrids compared to conventional hybrids. Kenyan farmers interviewed during the participatory evaluation of these trials could observe the differences in tassel and pollen formation but favored FNP hybrids overall due to the improved ear size and increased yield. As the technology broadens to other African countries it will be important to continue to seek farmer feedback on FNP hybrids. Yield improvement was correlated with reduced tassel size prior to anthesis and lack of production of pollen, as the formation of tassel structure and pollen competes for resources with grain production. Reducing this competition also reduces anthesis silking interval (ASI) under stress[29]. Reduced ASI has also occurred during selection for yield in SSA[26]. In FNP hybrids, 50% of the plants do not produce pollen, and partitioning of resources within the plant early on in development is shifted from the tassel in favor of the ear, leading to earlier silk protrusion and reducing ASI under stress. This change in partitioning results in more efficient use of nitrogen, a scarce resource for many smallholder farmers. Therefore, an added benefit of FNP hybrids is that they do not increase total N uptake but improve nitrogen utilization efficiency by reducing partitioning from the tassel in favor of the ear, increasing kernels per ear and kernel weight[13]. The adoption of modern FNP hybrids and the realization of associated yield benefits will still require nutrient inputs, given that current production largely relies on mining of nutrients which is unsustainable[30,31].

Widespread acceptance of FNP hybrids will be dependent on adoption by both farmers and seed companies. In this paper, we have described yield benefits to the farmer (about 200 kg ha$^{-1}$) and highlighted farmer acceptance, indicating the hybrids are likely to be adopted quickly. Under the conservative scenario (adoption of FNP in 10% of current maize hybrid area), we estimate that FNP would increase maize production in Africa by 0.245 Mt per year, valued at $40M. While this increase is relatively modest, for example when compared to the potential benefits of Bt maize[32], the benefits would still outweigh the cost by more than 6:1, indicating a good return to the research investment.

Apart from benefits to farmers, the technology also provides benefits to seed companies. These include: a reduction in detasseling costs, as the NPP females will not require detasseling during seed production; improved seed purity, as there is no self-pollination of female plants during seed production; and increased kernel numbers, leading to reduced seed production costs. Kernel number was increased by 6% in NPP plants under low N in these studies but was

**Table 2 Farmers prefer 50% non-pollen producing (FNP) hybrids over conventional (PP) hybrids.**

| Year, Season | Trait | Pollen producing (PP) | | Fifty-percent non-pollen producing (FNP) | | P-value |
|---|---|---|---|---|---|---|
| | | Mean | N | Mean | N | |
| 2017, Mid-season | Pollen shed | 3.90 ± 0.039 | 752 | 3.76 ± 0.039 | 752 | 0.001 |
| | Tassel formation | 3.98 ± 0.031 | 1065 | 3.80 ± 0.033 | 1065 | <0.0001 |
| | Yield | 3.49 ± 0.024 | 2134 | 3.63 ± 0.023 | 2134 | <0.0001 |
| | Overall evaluation | 3.60 ± 0.024 | 2134 | 3.68 ± 0.023 | 2134 | 0.003 |
| 2017, End-season | Tassel formation | 3.53 ± 0.019 | 2977 | 3.56 ± 0.019 | 2977 | 0.060 |
| | Yield | 3.49 ± 0.014 | 5839 | 3.62 ± 0.014 | 5839 | <0.0001 |
| | Overall evaluation | 3.49 ± 0.014 | 5807 | 3.61 ± 0.014 | 5807 | <0.0001 |
| 2018, Mid-season | Pollen shed | 3.52 ± 0.020 | 3222 | 3.53 ± 0.020 | 3221 | 0.462 |
| | Tassel formation | 3.48 ± 0.020 | 3222 | 3.54 ± 0.020 | 3221 | 0.012 |
| | Yield | 3.34 ± 0.014 | 6609 | 3.48 ± 0.014 | 6606 | <0.0001 |
| | Overall evaluation | 3.39 ± 0.014 | 6558 | 3.50 ± 0.014 | 6561 | <0.0001 |
| 2018, End-season | Tassel formation | 3.67 ± 0.013 | 7062 | 3.68 ± 0.013 | 7058 | 0.453 |
| | Yield | 3.54 ± 0.013 | 7109 | 3.64 ± 0.013 | 7111 | <0.0001 |
| | Overall evaluation | 3.58 ± 0.013 | 7062 | 3.66 ± 0.013 | 7058 | <0.0001 |

Farmer evaluations of pollen-producing (PP) and 50% non-pollen producing (FNP) hybrids in 2017 and 2018, on a 5-point hedonic scale (1 = dislike very much, 2 = like, 3 = neither like nor dislike, 4 = like, 5 = like very much) in mid-season and end-season for different criteria and overall. Values presented are mean ± s.e., N (number of data points) and P-values using pairwise t-tests. The number and description of participants in the questionnaires are shown in Supplementary Table 2.

not measured under favorable conditions more typical of seed production. In US trials, kernel number was increased by 9.6% in plants with NPP tassels compared with wild-type controls under optimum conditions[13]. This will be evaluated in African germplasm under seed production practices in SSA, but the expected increase in seed production of about 10% is an additional anticipated benefit to seed companies. The benefits to seed companies are also expected to help catalyze a shift towards more modern hybrids, improving the selection and purity of climate-smart hybrids available to smallholder farmers by providing incentives for seed companies to replace older, lower-yielding varieties with more recent higher-yielding ones. The replacement of older hybrids in the market will have added benefits for farmers, on top of those predicted from the ~200 kg ha$^{-1}$ benefit of growing FNP hybrids. We plan to collect additional data on the average age of hybrids that will be replaced, but assuming the average age of replacement is 10 years, this would reflect an additional 275 kg ha$^{-1}$ benefit to the farmer using conservative estimates of genetic gain. Therefore, the adoption of FNP hybrids would benefit farmers growing at the 2 Mg ha$^{-1}$ yield level by almost 25%, or 0.5 Mg ha$^{-1}$, approximately \$76 ha$^{-1}$ in added income.

The development and use of dominant male sterile technology across crops such as maize, rice, and wheat have been demonstrated but not yet widely applied, showing promise in delivering improved seed production and yield[33]. The Ms44-SPT system provides a unique opportunity to transform the maize hybrid seed industry in Africa, providing recognizable benefits to both seed companies and farmers. The FNP trait delivered using the Ms44-SPT system can deliver economic benefit in the form of improved input use efficiency to smallholder maize farmers faced both with limited ability to purchase recommended quantity of fertilizer and the uncertainty of drought stress.

## Methods

**Germplasm and incorporation of Ms44.** The dominant male-sterile allele *Ms44* was introgressed into five inbred maize lines. For the first year of trials, four of these inbreds were used, one had been backcrossed six times (BC$_6$) and three had been backcrossed four times (BC$_4$) to the respective recurrent parents. For each line conversion, four to five ear sources were selected for increase upon heterozygous marker calls for the donor allele and minimum introgression segment size. For subsequent trials, all five inbreds utilized had been backcrossed five or more times. As a dominant male-sterile allele, *Ms44* must be maintained in the heterozygous condition during increase by placing pollen from male-fertile plants onto silks of male-sterile plants. At each generation, the progeny rows segregate 1:1 for male-sterility. The five converted *Ms44* inbred lines were used as female parents and

crossed with 3–4 male inbred parental lines each to produce 18 unique single cross hybrid pairs. Female rows were segregated for pollen-producing (PP) and non-pollen producing (NPP) plants and these were classified and tagged separately at flowering. Ears were harvested separately for PP and NPP plants. The F1 hybrid seed harvested from NPP plants segregated 1:1 for pollen-producing (PP) and non-pollen producing (NPP) are referred to as 50% non-pollen-producing (FNP) hybrids. The F1 hybrid seed harvested from PP female plants produced 100% PP near-isogenic control hybrids. Eight three-way cross hybrids were produced by planting F1 seed harvested from NPP plants and crossing these to inbred PP males, resulting in three-way crosses segregating 1:1 PP and NPP plus the 100% PP controls.

**Yield testing.** From 2017 to 2019, yield trials were planted both on-station (OST) and researcher-managed on-farm (OFT) in Kenya, South Africa, and Zimbabwe (Supplementary Fig. 1 and Supplementary Table 1). Trials at experimental stations were conducted under optimal, low-N, heat, and drought stress. Optimal, heat, and drought stress sites were optimally fertilized based on local recommendations and received recommended weed and insect control measures. Optimal trials were planted during the main maize growing seasons, irrigated twice at planting and emergence, and supplemental irrigation was applied as needed to avoid drought stress. Managed drought trials were planted in the dry season and irrigation was withheld approximately 2 weeks prior to mid-anthesis. Delayed planting in the dry season allowed for high temperatures at the reproductive stage for heat stress trials. In low-N, fields had been depleted of nitrogen for at least 4-seasons. Rescue irrigation was only applied to avoid total crop loss when required. Depletion was achieved by applying no N fertilizer to plots and removing stover from the field after the grain was harvested.

Experiments were in a randomized complete block with a split-plot restriction, where the hybrid background was the main plot treatment and trait (PP or FNP) was the sub-plot treatment. Different hybrid combinations were grown in different years and locations depending on seed availability. On-station trials were 2–4 row plots of 5 m length and 0.75 cm between rows. There were 4–6 reps per location and usually more hybrid pedigrees planted across fewer locations. At selected OST locations, plants in the middle two rows were tagged at flowering according to phenotype: NPP for non-pollen-producing and PP for pollen-producing. When all PP plants were shed, the tassels from two PP and two NPP tagged plants in each plot were removed and the number of tassel branches was recorded. The tassel was cut at one inch above the flag leaf, oven dried to zero moisture and dry weight recorded. At 2–3 weeks after flowering, ear height (from the ground surface to the highest ear node) and plant height (to the tip of the tassel) were recorded for 4 PP and 4 NPP plants per plot. At OST locations in Zimbabwe, ear photos were taken and images analyzed[24] to estimate ear length, kernel number, 100 kernel weight, and grain weight per plant. Photos were taken using a tripod with the camera fixed at least 50 cm above the ears. Dehusked ears were placed on a black background, with 15–20 ears per photo. A 30 cm ruler was placed in the same orientation as the ears to be used as a reference.

For on-farm trials, smallholder farmers were identified by agricultural extension agents in each country. Extension agents were given a small monetary amount to cover all expenses related to trials. In Zimbabwe, additional seeds and inputs were given as compensation to farmers. On-farm trials were 2–4 row plots, 5 m rows with 0.75 m between rows. Plots were double planted and thinned, leaving an intra-

row spacing of 25 cm. There were 2 reps per location and multiple locations per year. In each country, project partners worked alongside extension agents and directly with farmers. Researcher-managed trials implemented by farmers are often higher yielding than farmers' own fields[27], thus farmers were asked to use appropriate pest and weed management, but not to apply N fertiliser. Target yields were less than 4 t ha$^{-1}$, based on the average yield of target farmers. Harvesting was conducted by hand, ears were shelled and grain weight and moisture were recorded. The yield on an area basis was calculated and adjusted to 155 g kg$^{-1}$ moisture.

Analysis was conducted using ASREML (VSN International Ltd). In the analysis for grain yield, the main effect of trait is considered as fixed effects and hybrid background and interaction between trait and hybrid background are treated as random effects. Location and interaction between location and trait are considered fixed. The blocking factors such as replicates are considered random. Yield for trait within hybrid was predicted using best linear unbiased predictor (BLUP), as the hybrid effect was treated as random. Yield for trait across hybrids was predicted using best linear unbiased estimates (BLUE), trait is considered a fixed effect. Differences between the 100% PP and the FNP trait were assessed by a two-sided $t$-test using the standard error of difference (SED) from the linear mixed model and were considered significant at the 5% confidence level.

**Farmer evaluations**. Farmer evaluations were organized in eight trial sites in Kenya in the main season of 2017 and 2018. The original sites were randomly selected from the trial sites in 2017. In 2018, two of the sites were dropped from the trials, so for farmer evaluations, they were replaced by nearby suitable sites. The evaluations were conducted twice in each year/season, mid-season (June–July) and end season (July–August). While breeders observed yield and other traits in the field, social scientists invited farmers to come and evaluate the entries in a subset of trials. The evaluations were double-blind: plots were identified by number and neither farmers nor facilitators/enumerators knew the treatments. For the participatory evaluations in 2017, 8 OFT sites were randomly selected from the trial sites, 4 in Central Kenya and 4 in Western Kenya. In 2018, 2 sites in Central Kenya were replaced, and the other 6 were maintained.

At each site, neighboring farmers were identified through farmer groups, local administration, and extension officers, and invited to evaluate the trials. In Kenya, women often tend to the farms while men are more likely to look for employment elsewhere, and it is common to have more female farmers participate[34]. The participants, 2697 in total, of which 62% were women, were adults of all ages (from 17 to 88) (Supplementary Table 2). Most participants were experienced farmers, with an average of 17 years of farming experience. Most had also finished primary education, with on average eight years of formal education. Most participants owned their farm, with an average size of almost 1 ha (0.85), more than half of which (0.5 ha) was planted in maize. Most participants practice a mixed crop/livestock system, with about two-thirds owning cattle and a quarter of oxen. The average cash income over the previous year was KES 92,617 (almost $1000), of which about half came from agriculture.

### Procedure

*Ethical compliance*. We have complied with all relevant ethical regulations regarding human research participants. The study protocols were approved by the Social Economics and Global Maize Programs of the International Maize and Wheat Improvement Center (CIMMYT). Participation of local partner organizations on the ethics review committee had not been implemented at the time the study was reviewed. Risk management plans for the health, safety and security of researchers were overseen by KALRO, ARC, and CIMMYT. Health and safety standards met or exceeded local requirements. Security measures followed guidance from the United Nations concerning staff residing and operating locally. Informed consent was obtained from all farmer participants in the preference survey conduct.

Farmers' evaluation of new technologies, including varieties, is a two-step procedure, where first the selection criteria or traits important to farmers are identified, followed by an evaluation of the new technologies or varieties on those criteria. Criteria during the first year were set and, based on discussions with farmers, four more criteria were added in 2018 (Supplementary Table 3)[35]. To confirm the importance of these criteria to the participants of this study, we asked them, individually, to give these a score for importance (0 = not important, 1 = somewhat important, 2 = important, 3 = very important) (Supplementary Table 3).

Participants were asked to evaluate the different entries on these criteria. In 2017, they evaluated the eight entries and two reps, so all 16 plots in total. In 2018 there were 16 entries and the participants only evaluated one of the two reps each, randomly assigned. To score the entries, they used a 5-point hedonic scale, following previous experience[36]. Experience has shown that using numbers for the scores can be confusing, as "1" can indicate both a very good or a very poor score. Therefore, letter scores were used, which correspond to the Kenyan school system and hence are easy for farmers to understand. The options were A (like very much), B (like), C (neither like nor dislike), D (dislike), and E (dislike very much)[34]. In 2017, farmers were randomly assigned to the control (without evaluations of tassel or pollen), treatment 1 (including the criterion "good tassel formation"), or treatment 2 (including both the tassel criterion and the criterion "amount of pollen shed"). As

the results of 2017 indicated treatments 1 and 2 were very similar, they were merged in 2018, with only one treatment group, whose members evaluated the entries on tassel and pollen. All criteria were expressed in both English and Kiswahili on the questionnaire, the national languages in Kenya. In the different counties, depending on the situation, the criteria were translated into local languages.

To analyze the scores, the alphabetical scores were converted to numerical scores (from A = 5 to E = 1), mean scores were calculated for all criteria and the mean scores for FNP and PP hybrids compared through pairwise $t$-test.

*Impact assessment*. To estimate maize area and production in SSA we used the FAOSTAT data from 2018 which includes 50 countries) with an area of 37.55 Mha a production of 70.51 Mtonnes and an average yield of 1.92 t ha$^{-1}$ [37]. For levels of adoption of improved maize varieties and hybrids we searched the literature and found data from the top 25 countries[38–44]. These 25 countries, including all countries with a maize area of more than 100 kha (except for Burundi and South Sudan) (Supplementary Table 4) plant 36.6 Mha (97.4% of maize area in SSA) with a production of 70.4 Mt. Multiplying adoption rates of improved maize varieties by country with their 2018 maize area, resulting in an estimated total area in improved maize varieties of 19.3 Mha (52.6%). Similarly, multiplying the % of hybrids for each country by the maize area led to an estimated area planted in hybrids in these countries at 12.6 Mha (34%). The yield benefit for each country was estimated using the regression from Fig. 1 ($\Delta$y = 0.006$x$ + 180.2). The weighted average (for the 25 countries with adoption figures, and area in hybrids used as weight) comes to 193 kg ha$^{-1}$.

To compare the benefits to the cost, we use the following project performance parameters: net present value (NPV), internal rate of return (IRR), and benefit–cost ratio (BCR)[25] (Supplementary Table 5). The cost of the development of the technology is estimated by the annual cost of the FNP project, US$ 1 million per year from 2010 to 2016 and US$1.6 million from 2017 to 2020. For the future, we expect the further development cost to be about $1.25 million per year from 2021 to 2024, after which the cost will gradually reduce from $0.8 million in 2025 to 0.1 in 2028. For the benefits, we assume the technology to be on the market in 2023, and to take 10 years to reach the target 10% adoption (market penetration of FNP hybrids as a percentage of hybrid seed), keeping maize production constant.

**Statistics and reproducibility**. OST was run in 11 locations across three countries between 2016 and 2019, on replicated plots for 26 genotypes on a total of 2784 plots. OFT was run in 79 farmer fields for 3 years, on replicated plots for 19 genotypes 1851 plots. Farmer evaluations were conducted on a total of 2697 farmers in eight farmer fields in 2017 and six farmer fields in 2018. Field data was analyzed using ASREML. Charts were produced in Excel and using packages ggplot2.

**Inclusion**. Research herein reported was designed and implemented with the full partnership of local researchers from KALRO, ARC, and CIMMYT. Five of the nine authors are local scientists. Data ownership and intellectual property rights are guided by the research collaboration agreement between the four implementing institutions. Research is locally relevant as determined in collaboration with local partners. Roles and responsibilities were co-developed and agreed upon prior to each season with extensive input and guidance from local partners. Capacity development was a critical component of both the SPTA project and the predecessor project Improved Maize for African Soils. Participation of international scientists supported by these projects in regional training events hosted by CIMMYT occurs annually. Four local scientists have participated in the projects as part of their dissertation research.

**Reporting summary**. Further information on research design is available in the Nature Research Reporting Summary linked to this article.

### Data availability

The data that support the findings of this study are available at https://data.cimmyt.org/dataset.xhtml?persistentId=hdl:11529/10548515 Figs. 1 and 2 and Table 2 have associated raw data; no personally identifiable information is included in these data sets. All other data are available from the corresponding author on reasonable request.

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

## Acknowledgements

This work was supported by the Bill & Melinda Gates Foundation project Seed Production Technology for Africa (grant number OPP1137722 and INV-018951), and the CGIAR Research Program on Maize (MAIZE). The CGIAR Research Program MAIZE receives W1&W2 support from the Governments of Australia, Belgium, Canada, China, France, India, Japan, Korea, Mexico, Netherlands, New Zealand, Norway, Sweden, Switzerland, the UK, the US, and the World Bank. Under the grant conditions of the Foundation, a Creative Commons Attribution 4.0 Generic License has already been assigned to the Author Accepted Manuscript version that might arise from this submission. We also recognize the broad contributions of CIMMYT, Corteva Agriscience, KALRO, and ARC as collaborators in the project. We acknowledge our colleagues at the research stations, farmers, and extension agents in Kenya, South Africa, and Zimbabwe who conducted yield trials and assisted with seed shipments.

## Author contributions

M.O. led and conceptualized the study with inputs from D.L., K.M., S.C., J.E.C., and M.A. E.H., J.E.C., D.L., and K.M. led the field trials. H.D.G. and M.N. led the farmer preference study. All authors contributed to the interpretation of the results and manuscript revisions.

## Competing interests

All authors declare that they have no known competing interests or personal relationships that could have appeared to influence the work reported in this paper. S.C. and M.A. are employees of Corteva Agriscience. Corteva Agriscience owns the rights to the technology. There is no competing interests as Corteva Agriscience is providing the technology royalty-free to licensed seed companies producing seed for smallholders in the region under the terms of the Seed Production Technology for Africa agreement (https://www.cimmyt.org/content/uploads/2019/03/CIMMYT-SPTA-project-brief-2020-07-web.pdf).
