## [Peer Review File · Communications Biology]

Reviewers' comments:

Reviewer #1 (Remarks to the Author):

General comments and summary: The manuscript reported a three-way hybrid seed production technology based on the Ms44-SPT system, which will benefit seed companies and farmers, as it eliminates the need for detasseling maize hybrid production fields during both hybridization steps. The resulting hybrids segregate 1:1 for pollen production, conserving resources for grain production and conferring a 200 kg ha⁻¹ benefit across a range of yield levels, which is equivalent to approximately six years of progress in plant breeding. Notably, the authors conducted largely on-farm trials with primary beneficiaries and evaluated the farmer perceptions of FNP hybrids. Generally, this manuscript provided some interesting primary data for increasing the hybrid maize yield in low input African farming systems based on the dominant male-sterility system. However, the manuscript still needs to be revised carefully, and more experiment data should be provided before it can be considered to be published in Communications Biology.

Main comments:

1. In Page 3 lines 71-72, the authors described "Finally, we investigate the mechanisms for the yield increase associated with the FNP trait". However, there is not any new experimental data on deciphering the mechanisms for the yield increase associated with the FNP trait in the manuscript, although the authors discussed this later based on the reported data.

2. In this manuscript, the three-way hybrid seed production technology is mainly based on the maize Ms44-SPT system, which has been reported previously (Fox et al., 2017). What is the novelty of this study? Although there are plenty of field trial data supporting the idea that "FNP hybrids have increased yield relative to PP hybrids".

3. Besides the Ms44-SPT system, other dominant male-sterility (DMS) system has also been developed in maize and rice, such as the DMS system based on the conserved function of p5126-ZmMs7 in maize and rice (An et al., 2020, PNAS), which may lead to the greatest increase of crop grain yield in the post-heterosis utilization era (Wan et al., 2021, Molecular plant). Thus, it is better to cite these works in this manuscript.

1) X. An, B. Ma, M. Duan, Z. Dong, R. Liu, D. Yuan, Q. Hou, S. Wu, D. Zhang, D. Liu et al, Molecular regulation of ZmMs7 required for maize male fertility and development of a dominant male-sterility system in multiple species, Proceedings of the National Academy of Sciences of the United States of America 117 (2020) 23499-23509.

2) X. Wan, X. Li, S. Wu, Breeding with Dominant Genic Male-Sterility Genes to Boost Crop Grain Yield in Post-Heterosis Utilization Era, Molecular Plant 14 (2021) 531-534.

Minor comments:

1. In Pages 8-9, Lines 168-170, "Based on FAO statistics and the adoption literature, the maize area in the top 25 maize producing countries is estimated at 36.6 M ha, of which 34.2% or 12.6 M ha is planted to hybrids." The information of "the top 25 maize producing countries" is not very clear here. Are these countries all belonged to Africa? Maybe it should be revised as "the top 25 maize producing countries in Africa".

2. In page 9 line 173, "and 17 a demand of > 100 tonnes" should be revised as "and 17 have a demand of > 100 tonnes".

Reviewer #2 (Remarks to the Author):

General comments

The manuscript provides a well-written and convincing summary of relevant results obtained from a field-based evaluation of a novel male sterility system for hybrid maize seed production using the dominant ms44 male sterility gene. The use of the ms44 system to produce an effective hybrid seed production system will be of broad interest to the international plant breeding and maize genetics communities.

The field-based evaluations using a combination of OST and OFT testing, in combination with the farmer-based evaluations of the hybrids and thus the hybrid technology, are of a sufficient scale (numbers of environments and hybrid backgrounds) that the conclusions drawn are well justified. Given the high importance of improving food security for many sub-Saharan countries of Africa the

manuscript will be of broad interest to the international agricultural scientific community. The studies that contributed to the manuscript are well designed, clearly explained and the statistical analyses based on mixed model methodology are robust. The inferences and conclusions drawn by the authors are clearly justified based on the results obtained from the experiments. In addition to demonstrating how the ms44 male sterility system is successfully applied to develop hybrid maize seed for a range of hybrid backgrounds, the study also demonstrates statistically significant yield advantages that are of a magnitude that can positively impact financial security of the intended target African farmers. The authors also provide some appropriate supporting results to explain how the yield benefit is mechanistically achieved. There is opportunity and value in a separate follow up manuscript to provide results further detailed investigations of the mechanistic basis of the documented yield improvements of the FNP hybrids over their paired PP versions.

Minor comments

Line 17: "removing" may be a better wording rather than "preventing".

Line 57: Use of NPP in the manuscript before acronym is defined a few lines later.

Line 314: check "(Fig. 5?)". Seems to be an editorial comment missed by the authors prior to submission.

General response

Reviewers' comments:

Reviewer #1 (Remarks to the Author):

General comments and summary: The manuscript reported a three-way hybrid seed production technology based on the Ms44-SPT system, which will benefit seed companies and farmers, as it eliminates the need for detasseling maize hybrid production fields during both hybridization steps. The resulting hybrids segregate 1:1 for pollen production, conserving resources for grain production and conferring a 200 kg ha⁻¹ benefit across a range of yield levels, which is equivalent to approximately six years of progress in plant breeding. Notably, the authors conducted largely on-farm trials with primary beneficiaries and evaluated the farmer perceptions of FNP hybrids.

Generally, this manuscript provided some interesting primary data for increasing the hybrid maize yield in low input African farming systems based on the dominant male-sterility system. However, the manuscript still needs to be revised carefully, and more experiment data should be provided before it can be considered to be published in Communications Biology.

Main comments:

1. In Page 3 lines 71-72, the authors described “Finally, we investigate the mechanisms for the yield increase associated with the FNP trait”. However, there is not any new experimental data on deciphering the mechanisms for the yield increase associated with the FNP trait in the manuscript, although the authors discussed this later based on the reported data.

Response: We have changed this sentence to “Additionally, we investigate changes in agronomic traits and yield components associated with the FNP trait” (Line 98-100 of revised manuscript). Our purpose was to characterize any differences of note in important traits and to understand changes in yield component traits rather than to try to understand the physiological mechanisms underlying any observed changes, and we have updated the language to reflect this.

2. In this manuscript, the three-way hybrid seed production technology is mainly based on the maize Ms44-SPT system, which has been reported previously (Fox et al., 2017). What is the novelty of this study? Although there are plenty of field trial data supporting the idea that “FNP hybrids have increased yield relative to PP hybrids”.

Response: There are a number of public-private-partnerships to deliver promising traits from research conducted in high income countries to improve yields in sub-Saharan Africa, but few examples of commercial success with impact for smallholder farmers. A major hurdle in translational research is demonstration of efficacy in the target population of environments where yields are very low and variable. Additionally, initial research was conducted in temperate maize backgrounds and it was important to demonstrate efficacy in African germplasm adapted to the target population of environments and in a larger number of genetic backgrounds to evaluate consistency of the effect across a broad array of genotypes which can serve a diversity of farmers across a wide geographical area. To be scalable, the trait must show repeatable benefits across a range of environments and genetic backgrounds. While this study is not novel in terms of the physiological mechanisms associated with the yield benefits of Ms44 derived hybrids, we feel that the demonstration of a significant yield benefit of a single-gene trait in evaluated in a large number of smallholder farmers' fields in three countries across a range of genetic backgrounds is an important translational research result. We also note that the current study reports on farmer preferences which indicate that our target beneficiaries recognize and appreciate the value of the FNP trait. Laajaj et al. (2020) highlighted the disparity

between researcher-managed and farmer-managed agronomic outcomes, with researcher management protocols differing from the realities of farmer management practices. In Zimbabwe, on-farm yields of more than 4 t ha⁻¹ are regularly reported (Setimela et al. 2016, 2017, 2018) yet national yields have remained below 1 t ha⁻¹ for over a decade. We ensured yields in this study were representative of our target beneficiaries. We agree we did not significantly highlight the novelty of this study and have since changed the text to reflect this. (Abstract lines 23-26; Introduction lines 74-83; Discussion lines 195-197)

3. Besides the Ms44-SPT system, other dominant male-sterility (DMS) system has also been developed in maize and rice, such as the DMS system based on the conserved function of p5126-ZmMs7 in maize and rice (An et al., 2020, PNAS), which may lead to the greatest increase of crop grain yield in the post-heterosis utilization era (Wan et al., 2021, Molecular plant). Thus, it is better to cite these works in this manuscript.

Response: We have added the review of Wan et al. (2021) (Lines 256-258). The publication of An et al. (2020) is an excellent detailed description of the mode of action of ZmMs7. We added reference to An et al. (2020) (Lines 69-72). The authors of the *ZmMs7* paper hypothesize that a dominant male sterile system could deliver a yield benefit to farmers citing Fox et al (2017) but they do not provide field data using the *ZmMs7* based system to support the claim.

Minor comments:

1. In Pages 8-9, Lines 168-170, "Based on FAO statistics and the adoption literature, the maize area in the top 25 maize producing countries is estimated at 36.6 M ha, of which 34.2% or 12.6 M ha is planted to hybrids." The information of "the top 25 maize producing countries" is not very clear here. Are these countries all belonged to Africa? Maybe it should be revised as "the top 25 maize producing countries in Africa".

Response: We apologies for this and have changed as suggested (Page 8, line 175 in revised version)

2. In page 9 line 173, "and 17 a demand of > 100 tonnes" should be revised as "and 17 have a demand of > 100 tonnes".

Response: We have changed as suggested. (Page 9, line 179 in revised version)

Reviewer #2 (Remarks to the Author):

General comments

The manuscript provides a well-written and convincing summary of relevant results obtained from a field-based evaluation of a novel male sterility system for hybrid maize seed production using the dominant ms44 male sterility gene. The use of the ms44 system to produce an effective hybrid seed production system will be of broad interest to the international plant breeding and maize genetics communities.

The field-based evaluations using a combination of OST and OFT testing, in combination with the farmer-based evaluations of the hybrids and thus the hybrid technology, are of a sufficient scale (numbers of environments and hybrid backgrounds) that the conclusions drawn are well justified. Given the high importance of improving food security for many sub-Saharan countries of Africa the manuscript will be of broad interest to the international agricultural scientific community.

The studies that contributed to the manuscript are well designed, clearly explained and the statistical analyses based on mixed model methodology are robust. The inferences and conclusions drawn by the

authors are clearly justified based on the results obtained from the experiments.

In addition to demonstrating how the ms44 male sterility system is successfully applied to develop hybrid maize seed for a range of hybrid backgrounds, the study also demonstrates statistically significant yield advantages that are of a magnitude that can positively impact financial security of the intended target African farmers.

The authors also provide some appropriate supporting results to explain how the yield benefit is mechanistically achieved. There is opportunity and value in a separate follow up manuscript to provide results further detailed investigations of the mechanistic basis of the documented yield improvements of the FNP hybrids over their paired PP versions.

Minor comments

Line 17: “removing” may be a better wording rather than “preventing”. **Response: Changed “reduces the cost of seed production by preventing the need for detasseling” to “improves efficiency and integrity of seed production by removing the need for detasseling.”** (Lines 18 and 19 of revised manuscript)

Line 57: Use of NPP in the manuscript before acronym is defined a few lines later. **Response: Changed as suggested (Line 63 in revised manuscript).**

Line 314: check “(Fig. 5?)”. Seems to be an editorial comment missed by the authors prior to submission. **Response: Apologies, changed as suggested (Line 317)**

Authors note on revised version relative to Table 1B. The number of data points involved in the measure of the traits in table 1B was inaccurate in the original submission and has been corrected. No other changes to the table were made. This was a transcription error when creating the table for the original submitted version and the correction does not reflect a re-analysis, therefore the reported P values are unchanged.

Trait	Pollen Producing (PP)	Fifty-percent non-pollen producing (FNP)	Difference	Change (%)	Pvalue	N
A						
Yield (kg ha ⁻¹)	3916.5 ± 73.2	4118.6 ± 73.3	202.1	5.2	<0.0001	4585
MST (%)	18.18 ± 0.25	18.22 ± 0.25	0.03	0.20	0.62	3923
B						
Ear height (m)	1.01 ± 0.01	1.00 ± 0.01	-0.01	-1.1	<0.01	471
Plant height (m)	1.93 ± 0.01	1.86 ± 0.01	-0.07	-3.8	<0.0001	469
Grain weight (g)	87.6 ± 3.7	95.2 ± 3.7	7.6	8.7	<0.0001	459
Tassel branch number	16.7 ± 0.18	15.3 ± 0.18	-1.4	-8.5	<0.0001	475
Tassel weight (g)	4.0 ± 0.09	3.73 ± 0.09	-0.27	-6.7	<0.001	475
C						
Kernel number	281.3 ± 3.7	297.9 ± 3.7	16.6	5.9	<0.001	464
Grain weight (g)	89.6 ± 1.4	95.1 ± 1.4	5.6	6.2	<0.001	464
100 kernel	31.7 ± 0.34	32.0 ± 0.34	0.3	0.9	<0.01	469

weight (g)						
Cob length (cm)	13.2 ± 0.18	13.9 ± 0.18	0.64	4.9	<0.0001	469

REVIEWERS' COMMENTS:

Reviewer #1 (Remarks to the Author):

The manuscript has been revised carefully according to my comments. I have no more comments on the revised version. It can be considered to be published in Communications Biology now.

Reviewer #2 (Remarks to the Author):

The author responses to the two earlier reviews of the manuscript clarify the points the authors intended and improve the manuscript. The revised version of the manuscript provides a convincing demonstration of the benefits for the SPT technology based on the dominant male sterility gene Ms44.

The combination of the yield results from on-station and the on-farm testing at relevant input and production yield levels together with the farmer preference studies in a range of relevant genetic backgrounds demonstrates a significant and meaningful yield improvement can be delivered to farmers in their on-farm conditions with the relevant input levels and that the hybrids are highly likely to be accepted by the intended farmers in Africa.

The additional benefit for effectiveness and efficiency of hybrid seed production for the seed companies is important and noted.

The scale and design of the experiments and analysis methodology are appropriate for the study.